# Key Signaling in Alcohol-Associated Liver Disease: The Role of Bile Acids

**DOI:** 10.3390/cells11081374

**Published:** 2022-04-18

**Authors:** Grayson W. Way, Kaitlyn G. Jackson, Shreya R. Muscu, Huiping Zhou

**Affiliations:** 1Center for Clinical and Translational Research, Virginia Commonwealth University, Richmond, VA 23298, USA; waygw@vcu.edu; 2Department of Microbiology and Immunology, Virginia Commonwealth University, Richmond, VA 23298, USA; jacksonkg@vcu.edu (K.G.J.); muscusr@vcu.edu (S.R.M.); 3Central Virginia Veterans Healthcare System, Richmond, VA 23249, USA

**Keywords:** alcohol-associated liver disease, bile acids, ethanol, steatosis, steatohepatitis, cirrhosis

## Abstract

Alcohol-associated liver disease (ALD) is a spectrum of diseases, the onset and progression of which are due to chronic alcohol use. ALD ranges, by increasing severity, from hepatic steatosis to alcoholic hepatitis (AH) and alcohol-associated cirrhosis (AC), and in some cases, can lead to the development of hepatocellular carcinoma (HCC). ALD continues to be a significant health burden and is now the main cause of liver transplantations in the United States. ALD leads to biological, microbial, physical, metabolic, and inflammatory changes in patients that vary depending on disease severity. ALD deaths have been increasing in recent years and are projected to continue to increase. Current treatment centers focus on abstinence and symptom management, with little in the way of resolving disease progression. Due to the metabolic disruption and gut dysbiosis in ALD, bile acid (BA) signaling and metabolism are also notably affected and play a prominent role in disease progression in ALD, as well as other liver disease states, such as non-alcoholic fatty liver disease (NAFLD). In this review, we summarize the recent advances in the understanding of the mechanisms by which alcohol consumption induces hepatic injury and the role of BA-mediated signaling in the pathogenesis of ALD.

## 1. Introduction

The use of alcohol is estimated to date back to 8000 BC, with the earliest proof in the form of chemical analysis dating back to 7000 to 6600 BC [1]. Alcohol has been used as a medicine, ointment, and cleaning agent and has been indulged in over the centuries. Alcohol-associated liver disease (ALD), also previously known as alcoholic liver disease, is a clinical illness caused by excessive and/or chronic alcohol use that has significant health and economic impacts. ALD is associated with a marked increase in lipid droplets in hepatocytes. It becomes classified as alcoholic fatty liver or steatosis once greater than 5% of hepatocytes develop this fatty phenotype [2]. These lipid droplet accumulations can be classified as macrovesicular, large lipid droplets that displace the nucleus and organelles or microvesicular, smaller lipid droplets that do not displace the nucleus, which tend to be more common in ALD [2]. The amount of fat in the liver can be measured by a specialized ultrasound device called a FirboScan [3]. However, alcoholic steatosis is rarely diagnosed, as it is largely asymptomatic. The lack of early detection often allows for further progression along the ALD spectrum that encompasses simple steatosis, alcoholic hepatitis (AH), alcoholic liver fibrosis, and alcoholic cirrhosis (AC) and can ultimately develop into hepatocellular carcinoma (HCC) (Figure 1). The current standard treatment for ALD is (and has been for decades) abstinence, as well as dietary and lifestyle changes prior to cirrhosis and liver transplantation for end-stage treatment [4,5,6]. ALD is closely associated with metabolic, physiologic, and inflammatory changes; gut dysbiosis; and altered bile acid (BA) synthesis, recycling, and signaling [6,7,8,9,10,11]. Lipid and lipoprotein profiles have been shown to be significantly different in patients with AH vs. heavy drinkers and could be used in prognosis [12]. AH patients have worse liver function compared to never-decompensated ALD, with AH being associated with bilirubinostatsis, severe fibrosis, ductular reaction, and aberrant gene expression [13]. Gut dysbiosis leads to increased inflammation, as well as altered BA signaling, due to changes in intestinal microbial modifications of BAs and has been associated with ALD, non-alcoholic fatty liver disease (NAFLD), and autoimmune liver disease [14]. More recently, not only has bacterially associated gut dysbiosis been important, but fungal changes in the intestinal microbiome have shown associations with ALD as well [15]. BAs are not only important in absorption of cholesterol and lipids but also act as critical signaling molecules that regulate lipid and glucose metabolism, as well as immune response. Impaired BA homeostasis has been linked with ALD and cirrhosis [16,17,18,19,20].

## 2. Alcohol-Associated Liver Disease (ALD)

### 2.1. Prevalence and Burden

ALD results from chronic and excessive alcohol consumption, with onset risk in doses as little as 10 g of pure ethanol per day [21]. Previous reports have shown that ALD is a major cause of liver disease worldwide and has become the leading cause of liver transplantation in the United States, surpassing the hepatitis C virus due to treatment advances [22]. ALD accounts for 30% of all HCC cases and HCC-specific deaths worldwide [23,24]. A recent predictive modeling study published in the Lancet Public Health estimates that age-standardized deaths due to ALD are expected to increase from 8.23 per 100,000 person-years in 2019 to 15.20 per 100,000 person-years in 2040 and that 1,128,400 people will die between 2019 and 2040 [25]. ALD is the leading chronic cause of alcohol-attributable deaths, accounting for 18,164 deaths annually in the United States [26]. 

### 2.2. Alcohol Metabolism

Many of the toxic effects of ethanol are associated with and tied to its site of metabolism. Ethanol is readily absorbed in the gastrointestinal tract, with only 2 to 10% of the absorbed amount being eliminated through the lungs, urine, and sweat, whereas the remaining amount is oxidized, primarily in the liver [27]. Ethanol metabolism is performed by several enzymes. Ethanol is primarily oxidized by alcohol dehydrogenase (ADH) in hepatocytes to acetaldehyde. Acetaldehyde loses a hydrogen by aldehyde dehydrogenase (ALDH) and NAD^+^ to form acetate and NADH [1]. A lysine substitution for glutamate at position 504 of *ALDH2* leads to a near completely inactive form called *ALDH2*2* [1]. *ALDH2*2* leads to an alcohol flush reaction and is relatively common in people of Chinese, Japanese, and Korean descent but essentially absent from people of European or African ancestry [1]. Acetate can then be converted to acetyl-CoA, which can be oxidized in the tricarboxylic acid cycle. ADH and ALDH each require coenzyme nicotinamide adenine dinucleotide (NAD^+^) for the transfer of oxygen, reducing it to NADH [28,29,30,31]. There are multiple forms of ADH and ALDH, which are encoded by different genes. The genetic variants of ADH and ALDH can affect ethanol metabolism [28,31]. Previous research has focused on identifying which genes may be more strongly associated with alcoholism, with Edenberg summarizing that *ALDH2*2* may be protective against alcohol dependence [28]. Although ADH is the major enzyme in ethanol metabolism, there are two other well-known enzymes: cytochrome P450 2E1 (CYP2E1), part of the microsomal ethanol oxidizing system (MEOS); and catalase [27,31,32]. CYP2E1 participates in the metabolism of acetone and fatty acids and is typically found in microsomes or vesicles within the endoplasmic reticulum [29,31,32,33]. Although CYP2E1 is generally found in the liver, it can be found in multiple organs, and its expression is inducible [29,31,32,33]. Catalase, found in all major organs, is an antioxidant enzyme that is typically known for its role in converting hydrogen peroxide into water and molecular oxygen [27,31,32]. It has recently been shown that peroxisome proliferator-activated receptor α (PPARα) activation completely shifts ethanol metabolism from the reactive oxygen species (ROS)-generating CYP2E1 pathway to the ROS-scavenging catalase pathway and accelerated alcohol clearance [34]. Ethanol also induces cytochrome P450 2A5 (CYP2A5), and this induction is regulated by nuclear factor-erythroid 2-related factor 2 (NRF2) [35]. Whereas the primary site for alcohol metabolism is the liver, some metabolism may occur elsewhere and lead to some tissue damage in other sites. 

### 2.3. Alcohol Toxicity and ALD Pathology

Excessive alcohol use is the leading preventable cause of death in the United States, and chronic alcohol conditions account for 51,078 deaths per year [26]. Ethanol is a toxic compound in excessive and chronic amounts and leads to the development of ALD. Ethanol toxicity is primarily due to its metabolism, which explains why most damage is associated with the liver, the primary site of metabolism. However, ethanol itself can interact with membrane phospholipids, stimulate Kupffer cells, and increase oxidative stress [36]. When ethanol is ingested, it undergoes first-pass metabolism once encountering the gut wall and transits to liver hepatocytes via the hepatic portal vein system. At the hepatocytes, depending on the duration and amount of ethanol exposure, an increasing amount of hepatotoxicity results due to the production of acetaldehyde from ethanol by ADH, as well as decrease in the NAD^+^/NADH ratio and mitochondrial damage [37]. Acetaldehyde covalently binds to microtubules, leading to hepatocyte swelling, as it blocks the excretion of proteins [38]. Acetaldehyde can lead to a number of DNA-altering and -degrading effects, such as DNA adducts, point mutations, DNA crosslinking, and single- and double-strand breaks [39]. The increasing level of hepatotoxicity can acutely lead to AH or, in chronic cases, an increase in fatty hepatocytes and alcoholic steatosis (AS). The accumulation of fatty acids in hepatocytes seen in AS is due to alcohol’s effects on lipid metabolism through altered lipid uptake and export, de novo synthesis, fatty acid oxidation, droplet formation, and catabolism [40]. Alcohol increases adipocyte lipolysis, largely through insulin resistance, increasing circulating non-essential fatty acids, which leads to an increase in fatty acid uptake by hepatocytes [40,41]. Alcohol-associated hepatocyte fatty acid accumulation is also due to an increase in hepatic fatty acid transporters. CD36, also called fatty acid translocase, is a fatty acid transport protein that uptakes circulating fatty acids and is normally minimally expressed in hepatocytes; however, previous studies have shown that CD36 is highly inducible by alcohol, contributing to hepatic steatosis, and that CD36 ablation alleviates ethanol-induced hepatocyte lipid accumulation [42,43]. Alcohol has also been shown to alter several lipid regulatory factors, such as sterol regulatory element binding protein-1c (SREBP-1c), carbohydrate response element-binding protein (ChREBP), and PPARα [40]. The most significant cause of ALD hepatocellular fatty acid accumulation is ethanol inhibition of mitochondrial β-oxidation [40,44]. This inhibition results from a reduction in AMP-activated protein kinase (AMPK) activity, increasing acetyl-CoA carboxylase activity and increasing malonyl-CoA levels [45]. This leads to the inhibition of carnitine palmitoyltransferase 1, which is needed for fatty acid translocation for mitochondrial β-oxidation. As hepatocytes are continually exposed to ethanol, these toxic effects compound, causing chronic liver damage. This chronic liver damage leads to fibrosis by hepatic stellate cells (HSCs) switching from a quiescent state to an active state [46]. Once in an active state, HSCs begin depositing collagen in liver tissue [46]. Continued fibrosis leads to the accumulation of scar tissue on most of the liver, becoming AC. AC can ultimately result in HCC due to accumulated cellular and DNA damage.

Liver toxicity from ethanol can be impacted by several factors, such as diet, exercise, environment, and genetics. Recently, genetic analyses identified that genetic variants in phospholipase domain-containing protein 3 (PNPLA3) and haptoglobin increase the risk for AH risk based on increased total bilirubin and a model for end-stage liver disease score, which were used as surrogates for AH severity [47]. CYP2A5 expression has also been shown to be induced by ethanol, and CYP2A5^−/−^ mice develop more severe alcoholic fatty liver than wild-type mice, revealing a protective effect of CYP2A5 [35]. The same study also investigated CYP2A5’s protection and identified a relationship with the PPARα-fibroblast growth factor 21 (FGF21) axis [35]. Although the liver is the primary site of alcohol metabolism, its metabolically associated damage can also occur in other tissues, such as the brain, intestines, and cardiovascular system [19,29,48]. The reduction of ethanol to acetaldehyde and then acetate has been associated with the production of reactive oxygen species; the bioreactive aldehydes produced lead to neurotoxicity and neurodegeneration [29,49]. Excessive alcohol consumption can lead to alcoholic cardiomyopathy through a not yet fully elucidated mechanism that is believed to revolve around the alteration of the mitochondria, oxidative stress, and inducing apoptosis in cardiomyocytes [50,51,52,53]. Chronic intake of ethanol and its metabolites can also lead to gut dysbiosis. Continued alcohol use in cirrhotic patients results in significant duodenal, ileal, and colonic dysbiosis; higher endotoxemia; and higher systemic and ileal inflammatory expression [54]. However, at high levels of ethanol, extrahepatic metabolism of ethanol to acetaldehyde occurs by CYP2E1 and catalase. In the brain, BAs have been shown in previous studies to increase in patients with alcoholic steatohepatitis and in cirrhosis [55,56]. Cirrhosis is often associated with hepatic encephalopathy (HE). Comparing cirrhotic patients with HE to cirrhotic patients without HE revealed that HE patients had significantly worse systemic inflammation, gut dysbiosis, and hyperammonemia compared to controls and non-HE cirrhotic patients [57]. Liver cirrhosis also affects the bone marrow, with advanced cirrhotic patients having significantly reduced hematopoietic stem cells, mesenchymal stem cells, Schwann cells, neural fibers, and endothelial cells, as analyzed by immunohistochemistry [58].

## 3. Bile Acids

### 3.1. Bile Acid Enterohepatic Circulation

BAs are amphipathic steroid molecules derived from a multistep enzymatic pathway. De novo BA synthesis begins with cholesterol in the liver. After BAs are formed in the hepatocytes of the liver, they are transported into the bile canaliculi by the efflux transporter bile salt export pump (BSEP) and multidrug resistance-associated protein 2 (MRP2). Once in the bile canaliculi, BAs then flow to and are stored in the gallbladder. Upon ingestion of fats and proteins, cholecystokinin (CCK) is released from endocrine cells in the small intestine. CCK then signals the smooth muscle cells of the muscularis layer of the gallbladder to contract and the sphincter of Oddi to relax, releasing bile into the cystic duct. Bile enters the common bile duct from the cystic duct and flows into the ampulla of Vater before entering the duodenum. Most of the BAs are reabsorbed in the intestines via the hepatic portal vein to end up in the liver again and start the cycle anew. This cycle is referred to as enterohepatic circulation. A proportion of 95% of BAs are absorbed in the intestines, with the majority being absorbed in the terminal ileum by active transport via the apical sodium-dependent bile transporter. The BAs are then transported across the enterocyte into the sinusoidal membrane, where epithelial cells’ organic sulfate transporter-α and -β (OST-α and -β) transport the BAs into portal blood. Once BAs in the portal vein reach the liver, hepatocytes uptake the BAs via Na^+^-taurocholate cotransporting polypeptides (NTCPs) and organic anion transporting polypeptides (OATPs) on the basolateral membrane [59,60,61]. OATP1B1 and OATP1B3 have been shown to preferentially transport conjugated BAs over unconjugated BAs [62]. The BAs left in the intestinal lumen are altered by gut bacteria. Bacterial modifications include deconjugation, 7-dehydroxylation, amidation, oxidation-reduction, esterification, and desulfation [63,64,65]. Humans have a total bile production of ~600 mL and a BA pool size of 4 to 6 g, releasing approximately 12 to 30 g into the intestines daily; these BAs recirculate an average of three to five times [66,67,68,69]. BAs can be categorized as primary bile acids or secondary bile acids, each of which can be conjugated or unconjugated. In humans, the liver produces two primary bile acids—cholic acid (CA) and chenodeoxycholic acid (CDCA)—whereas rodents produce CA, CDCA, and α- and β-muricholic acids (α- and β-MCA). Human secondary bile acids consist of deoxycholic acid (DCA) and lithocholic acid (LCA), whereas mouse secondary bile acids consist of murideoxycholic acid (MDCA), hyodeoxycholic acid (HDCA), and ω-Muricholic acid (ω-MCA) [69]. Both humans and mice can have ursodeoxycholic acid (UDCA). BAs can be conjugated by the addition of taurine or glycine. Murine BAs are mostly taurine-conjugated, whereas human BAs are mainly glycine-conjugated [70]. Another distinguishing difference is that the murine BA pool is more hydrophilic than the human BA pool [71].

### 3.2. Bile Acid Metabolism

The primary bile acids are produced from cholesterol via two well-characterized pathways: the classical (neutral) pathway and the alternative (acidic) pathway [72]. The classical pathway accounts for ~90% of BA formation, with the alternative pathway making up the final 10% (Figure 2). This catabolic process requires more than a dozen enzymes to modify the cholesterol steroid core. Human BAs have 24 carbon atoms, with a steroid core that consists of three six-member rings and a five-member ring. The rate-limiting steps for both the classical and alternative pathways are the initial enzymes for each of them: cholesterol 7α-hydroxylase (CYP7A1) and cholesterol 27α-hydroxylase (CYP27A1), respectively. CYP7A1 hydroxylates at the C7 position of cholesterol to form 7α-hydroxycholesterol, whereas CYP27A1 hydroxylates cholesterol at C27 to form 27-hydroxycholesterol. In the classical pathway, 7α-hydroxycholesterol is then converted to 7α-hydroxy-4-cholesten-3-one (C4) by 3β-hydroxy-Δ^5^-C27-steroid oxidoreductase (HSD3B7) [73]. A multi-enzymatic process that results in a double-bond reduction, further hydroxylation, and side-chain cleavage converts C4 to CDCA via aldo-keto reductase 1D1 (AKR1D1), 3α-hydroxysteroid dehydrogenase (3αHSD), and CYP27A1 [73]. Microsomal sterol 12α-hydroxylase (CYP8B1) interacts with C4 to form 7α,12α-hydroxy-4-cholesten-3-one and regulates the CA-to-CDCA ratio [74]. 7α,12α-hydroxy-4-cholesten-3-one is then altered by other subsequent enzymes to form CA. In the alternative pathway, CYP27A1, located in the inner mitochondrial membrane, is the first step of the enzymatic process, followed by oxysterol 7α-hydroxylase (CYP7B1). A recent study showed that CYP7B1 is responsible for controlling the levels of intracellular regulatory oxysterols produced by the alternative pathway [75]. The alternative pathway is believed to mainly produce CDCA. Following further subsequent enzymatic alterations by AKR1D1 and 3αHSD, as well as others, CDCA is formed [73]. Bile acid–CoA synthase (BACS) and bile acid–CoA: amino acid N-acetyltransferase (BAAT) then add glycine or taurine to CA and CDCA to produce the conjugated bile acids glycocholic acid (GCA), taurocholic acid (TCA), glycochenodeoxycholic acid, and taurochenodeoxycholic acid (TCDCA). In mice, CDCA and UDCA are converted to α-MCA and β-MCA by cytochrome p450 2C70 (CYP2C70) and are then conjugated with taurine or glycine [71,76].

The bile acids then undergo enterohepatic circulation as previously described. In the intestine, BAs can be further modified by a variety of bacteria to form secondary BAs. One of the major bacterial alterations of BAs is the deconjugation of conjugated BAs by bile salt hydrolase (BSH) enzymes. BSH protein sequences have been identified in 591 intestinal bacteria strains within 117 genera of human microbiota and reclassified into 8 phylotypes [77]. *Lactobacillus* BSH has been shown to have the highest enzyme activity, whereas BSH phylotypes BSH-T5 and -T6 are mainly from *Bacteroides*, with a high percentage of paralogs that exhibit different enzyme activity [77]. Another important microbial modification is 7α-dehydroxylation, which converts CA and CDCA to DCA and LCA, respectively, as well as 7β-dehydroxylation, which converts UDCA to LCA [64]. BAs can also be modified by dehydrogenation, oxidation, epimerization; more recently, gut microbiota have been shown to conjugate amino acids to bile acids, termed microbially conjugated bile acids [78,79]. BA synthesis, reabsorption, metabolism, and the effects that BAs mediate are heavily regulated by and carried out through their interactions with receptors.

**Figure 2 cells-11-01374-f002:**
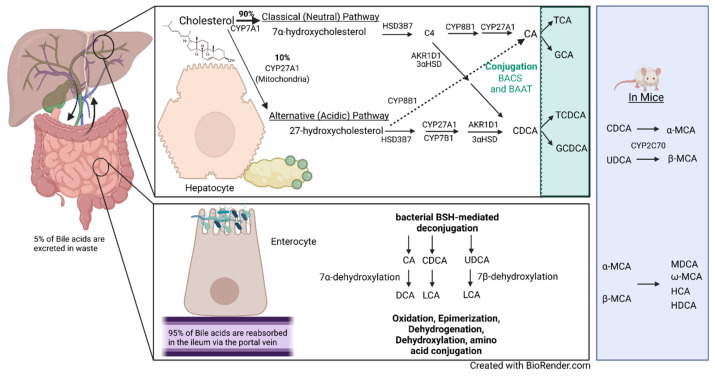
Bile acid metabolism and enterohepatic circulation. The human liver produces 0.5 g of bile per day via de novo synthesis. Only 5% of the secreted daily amount of bile acids is lost as waste each day. Bile acid synthesis can occur via the classical (neutral) pathway or the alternative (acidic) pathway. The classical pathway is regulated by cholesterol 7α-hydroxylase (CYP7A1), the first enzyme and rate-limiting step in the pathway, which converts cholesterol to 7α-hydroxycholesterol. CYP7A1 is located in the endoplasmic reticulum of hepatocytes. The alternative pathway’s first enzyme is sterol 27-hydroxylase (CYP27A1), present in macrophages and other cells, which converts cholesterol to 27-hydroxycholesterol and is the rate-limiting step of the alternative pathway. The classical pathway accounts for 90% of synthesized bile acids through a multi-enzymatic step. In the classical pathway, 7α-hydroxycholesterol is converted to 7α-hydroxy-4-cholesten-3-one (C4) by 3β-hydroxy-Δ^5^-C27-steroid oxidoreductase (HSD3B7). The classical pathway then converts C4 to cholic acid (CA) or chenodeoxycholic acid (CDCA) by a multi-enzymatic step that includes microsomal sterol 12α-hydroxylase (CYP8B1), aldo-keto reductase 1D1 (AKR1D1), 3α-hydroxysteroid dehydrogenase (3αHSD), and CYP27A1. The alternative pathway mainly produces CDCA in a multi-step enzymatic process that includes CYP27A1, oxysterol 7α-hydroxylase (CYP7B1), AKR1D1, and 3αHSD, among others. CA and CDCA are conjugated with glycine or taurine by bile acid–CoA synthase (BACS) and bile acid–CoA: amino acid N-acetyltransferase (BAAT) to produce the conjugated bile acids glycocholic acid (GCA), taurocholic acid (TCA), glycochenodeoxycholic acid, and taurochenodeoxycholic acid (TCDCA). Mice also convert CDCA and ursodeoxycholic acid (UDCA) to α-muricholic acid (α-MCA) and β-muricholic acid (β-MCA) by cytochrome p450 2C70 (CYP2C70), which can then be conjugated with glycine and taurine. In the intestines, bile acids are deconjugated by bacterial bile salt hydrolases (BSHs). CA and CDCA undergo 7α-dehydroxylation to form (deoxycholic acid) DCA and lithocholic acid (LCA), respectively. 7β-dehydroxylation converts UDCA to LCA. The murine-specific bile acids created are murideoxycholic acid (MDCA), ω-muricholic acid (ω-MCA), hyocholic acid (HCA), and hyodeoxycholic acid (HDCA). Bile acids are further modified by dehydrogenation, dehydroxylation, oxidation, and epimerization; more recently, gut microbiota have been shown to conjugate amino acids to bile acids, termed microbially conjugated bile acids [78].

### 3.3. Bile Acid Signaling

Maintaining BA homeostasis is an important physiological process that focuses on regulating synthesis, absorption, and excretion. This regulation is controlled by several specific nuclear and surface receptors, transporters, and their subsequent signaling cascades and secondary signaling molecules. BAs can interact with several receptors, leading to the activation of a plethora of secondary signaling molecules that lead to a variety of metabolic and homeostatic changes. BA nuclear receptors include the farnesoid X receptor (FXR) [80], the pregnane x receptor (PXR) [81,82], and the vitamin D receptor (VDR) [83]. The FXR has two members in mammals: FXRα and FXRβ [84]. FXRα has four isoforms, FXRα1-α4, whereas FXRβ encodes functional receptors in other species but is a pseudogene in humans and primates [85,86,87]. The FXRα isoforms exhibit locational differences in expression, with FXRα1 and FXRα2 being moderately expressed in the ileum and adrenal glands and FXRα3 and FXRα4 being highly expressed in the ileum and moderately expressed in the kidneys [87]. FXR binds to CDCA, LCA, DCA, and CA [80,88,89]. The FXR has been shown to regulate the metabolism of and is the major coordinator of bile acids, carbohydrates, lipids, and absorption of dietary fats and vitamins and plays an important role in the anti-inflammatory response and inhibition of hepatocarcinogenesis [84,90]. Upon activation, FXR forms a heterodimer with retinoid X receptor α (RXRα), the allosteric signal transduction of which was recently investigated, showing changes in affinity and conformational changes in helix 11 of the FXR (Figure 3) [91]. Activation and heterodimerization lead to the expression of small heterodimer partner (SHP), BACS, and BAAT and lead to transcriptional repression of CYP7A1 and liver homolog 1 [92,93,94]. SHP leads to repression of NTCP, reducing BA uptake of hepatocytes and CYP7A1, reducing BA synthesis [92,95]. FXR also leads to the expression of bile salt export pump (BSEP), OST-α and -β, multidrug resistance protein 2 (MRP2), and multidrug resistance protein 3 (MDR3) [96,97,98,99]. SHP mediates liver X receptor (LXR) anti-inflammatory effects by SUMOylation of LXR, with knockdown of SHP abrogating LXR SUMOylation, preventing its anti-inflammatory effects [100]. BSEP, MRP2, and MDR3 all transport their targets into the bile canaliculus and are critical in the healthy production and composition of bile. PXR is activated by 3-keto-LCA and LCA [82,89]. PXR activation is also associated with RXRα heterodimerization and leads to increased drug metabolism, drug transport, and lipogenesis while decreasing gluconeogenesis, glycogenolysis, β-oxidation/ketogenesis, and BA synthesis [89,101]. PXR has been shown to regulate liver size in mice by treatment of PXR-selective activators, leading to liver enlargement and induction of regenerative hybrid hepatocyte reprogramming via a Yes-associated protein mechanism [102]. VDR is activated by secondary bile acids, such as LCA and LCA derivatives, including LCA acetate and LCA propionate [83,103,104]. Like other BA nuclear receptors, upon activation, VDR is associated with RXR and then binds to specific DNA elements to affect various proteins at the transcriptional level [83]. VDR is highly regulated in the intestinal tract and is also expressed in the kidney [89]. VDR as a BA receptor may play a protective role. In intestinal cells, VDR induces expression of CYP3A, which metabolizes toxic LCA and can help prevent degradation of the intestinal barrier and entrance of LCA into enterohepatic circulation, leading to LCA hepatoxicity [83,105]. BA G protein-coupled receptors include Takeda G protein-coupled receptor 5 (TGR5), sphingosine-1-phosphate receptor 2 (S1PR2), and muscarinic acetylcholine receptor M3 (M3R). TGR5, also called the G protein-coupled bile receptor 1 (GPBAR1), which was the first BA non-nuclear receptor discovered, was also found to mediate a range of physiological functions, such as maintenance of metabolic homeostasis and insulin sensitivity [106,107]. TGR5 is mainly activated by unconjugated and secondary BAs [106]. TGR5 can associate with either stimulatory or inhibitory G alpha proteins (Gα_s_ or Gα_i_), depending on the cell type (Figure 3) [89]. In most cells, TGR5 couples with Gα_s_ and BA binding and leads to receptor internalization, activation of extracellular signal-related kinase, mitogen-activated protein kinase, and activation of adenylate cyclase, as well as an increase in cyclic AMP [108]. In cholangiocytes, TGR5 can couple with either Gα_s_ or Gα_i_, depending on subcellular localization [109]. In the primary cilium, TGR5 inhibits cell proliferation by coupling with Gα_i_; conversely, in the apical plasma membrane, TGR5 promotes cell proliferation by coupling with Gα_s_ [109]. In a Barrett’s esophageal adenocarcinoma cell line, TGR5 was observed to be coupled with Gα_q_ and Gα_i3_, but only Gα_q_ exhibited signal transduction after ligand binding [110]. SP1R2 is coupled with Gα_i_, Gα_q_, and Gα_12/13_ (Figure 3) [17,111]. Gα_i_ activates the phospholipase C(PLC)/IP3/DAG pathway, PI3K-Akt signaling pathway, and the MAPK pathways [112,113,114]. Gα_q_ solely activates the PLC/IP3/DAG pathway [112,113,114]. Gα_12/13_ activates the Rho/ROCK NF-κB and PTEN pathways, which lead to inflammation and stress fiber formation [112,113,114]. BA-induced activation of S1PR2 is mainly mediated by conjugated primary BAs and coupled with Gα_i_ [69]_._

### 3.4. Bile Acids in Disease

BA accumulation has been associated with liver injury, chronic liver disease, inflammation, and tumorigenesis (Figure 4) [20,115,116]. High levels of secondary bile acids in feces and blood have been associated with cholesterol gallstones and colon cancer [117]. An observational study revealed that BAs are significantly increased in liver cirrhosis, with the authors suggesting using total and individual BAs, especially primary CBAs, as non-invasive markers for diagnosis of liver cirrhosis, with potential for use as indicators for HCC [56]. In a recent study looking at BAs and cancer cachexia, mouse total BA levels significantly increased, but BA synthesis enzyme expressions were inhibited [118]. Changes in BA metabolism and an increase in BA conjugation in clinical patients were also observed [118]. CBA TCA has been shown to significantly promote cell proliferation, migration, invasion, transformation, and cancer stem cell expansion in esophageal adenocarcinoma cells via S1PR2 [119]. DCA dietary supplementation in a preclinical non-alcoholic steatohepatitis mouse model restored BA concentrations in portal blood; increased TGR5 and FXR signaling; ameliorated metabolic dysbiosis; and protected against steatosis, ballooning, and macrophage infiltration [120]. However, an abnormally high level of microbially modified DCA has been associated with gut dysbiosis, disruption of mucosal physical and functional barriers, and intestinal carcinogenesis [121,122]. Downregulation of FXR, the major BA nuclear receptor, alters the gut microbiome by facilitating *Bacteroides fragilis* colonization, which leads to the promotion of colorectal tumorigenesis [123]. Inhibition of FXR and BA metabolism modulation by trimethylamine N-oxide exacerbates steatosis in non-alcoholic fatty liver disease [124]. The FXR has also been investigated as a therapeutic target for cardiometabolic diseases [125]. Conjugated BAs can interact with S1PR2 and promote neuroinflammation during hepatic encephalopathy in mice, suggesting that reduction in BAs or S1PR2 signaling is a potential therapeutic strategy for hepatic encephalopathy [126]. BA activation of M3R has been shown to induce proliferation in human colon cancer cell lines via epidermal growth factor receptors, and M3R activation stimulates colon cancer cell invasion through MAPK-ERK1/2 and induction of matrix metalloproteinase-1 expression [127,128]. 

## 4. Bile Acids in ALD

Alcohol alters many metabolic pathways, including BA and cholesterol metabolism, and causes inflammation and injury to multiple organ systems. Conversely, maintaining metabolic homeostasis is protective against the deleterious effects of alcohol. The authors of one murine study observed that ethanol increased BA levels, BA synthesis genes (CYP7A1, CYP27A1, CYP8B1, and BAAT), and BA transporters but downregulation in BA transporter NTCP in the liver and nuclear receptor FXR in the ileum [129]. However, in humans, researchers observed that total and conjugated BAs are significantly increased in patients with AH, but de novo synthesis is suppressed based on a decrease in CYP7A1 gene expression and C4 serum levels [130]. This same study found that fibroblast growth factor 19 (FGF19) correlated with total and conjugated Bas, and FGF19 has significant associations with bilirubin and gamma-glutamyl transferase [130]. Plasma TCDCA and tauroursodeoxycholic acid levels have been observed to be directly related to disease severity in ALD, whereas fecal ursodeoxycholic acid was inversely related [116]. CYP7A1-deficient mice (the rate-limiting step in BA synthesis) have greater hepatic inflammation and injury from alcohol than wild-type mice, and hepatic injury is ameliorated in CYP7A1 transgenic mice, suggesting CYP7A1 and BA synthesis play a protective role in ALD [131]. An altered BA glycine-to-taurine ratio has been associated with stage-specific liver disease patterns and may be used as new biomarkers for monitoring disease progression [132]. PPARα has been found to be significantly reduced in the liver of severely alcoholic hepatitis patients [34]. The modulation of fatty acid and bile acid metabolism by PPARα showed protective effects against ALD via investigation of comparative gene expression of wild-type and PPARα-null mice [133]. As mentioned previously, PPARα also showed protective effects against ethanol metabolism toxicity by shunting it from the ROS-generating CYP2E1 pathway to the ROS-scavenging catalase pathway [34]. PPARα-null mice exhibited an increase in alcohol-associated accumulation of triglycerides, hepatic cholic acid and derivatives, and induction of fibrogenesis genes compared to wild-type mice [133]. The observed disparities contributed to PPARα’s mitochondrially protective effects via modulation of three mitochondrial metabolic pathways [133]. In preclinical studies, a PPARα agonist, seladelpar, was shown to reduce ethanol-induced liver disease through gut barrier restoration and bile acid homeostasis [134]. Coupling the preclinical protective effects of PPARα and previously observed results indicating that bile acids induce human PPARα via FXR activation [135] suggests that BAs play an important role in the management of ALD progression. PPARα has also been shown to regulate fibrate-mediated suppression of bile acid synthesis through downregulation of cholesterol 7α-hydroxylase and sterol 27-hydroxylase [136]. Mice deficient in BA receptor TGR5 had worse alcohol-associated injury than wild-type mice, with an increase in liver macrophage recruitment, altered bile acid profile, and gut microbiota dysbiosis that, when transplanted to WT mice, led to exacerbation of alcohol-induced inflammation [10]. Alcohol consumption induces a change in the gut microbiota, which leads to an increase in bacteria with choloylglycine hydrolase, a BSH, and was coupled with a lower secretion of fibroblast growth factor 15 [137]. Although deficiency or inhibition of FXR has been shown to alleviate obesity in NAFLD mice [138,139], in ALD, it has been shown to cause more damage, and FXR agonists improve ALD [140,141]. In mice who were either binge-fed or chronically given ethanol, treatment with a TGR5 or FXR agonist ameliorated liver inflammation, steatosis, and injury, which was associated with a reduction in the IL-1β pro-inflammatory cytokine [142]. This supports previous research suggesting that TGR5 activation in Kupffer cells leads to a decrease in the pro-inflammatory cytokines IL-1β and tumor necrosis factor-α [143]. Another beneficial effect of TGR5 and FXR agonism is the regulation of NLRP3 inflammasome through protein kinase A activation and ubiquitination of NLRP3 [142].

## 5. Conclusions

ALD is now the leading indication for liver transplantation in the United States of America [22]. Liver transplantation is still the gold-standard end-state treatment for ALD, with few changes in treatment options over the decades. Front-line treatment for ALD prior to fibrosis still consists of abstinence, as well as dietary and lifestyle modifications, with little in the way of pharmacotherapeutics. Recently, development and testing of therapeutic agents targeting various BA receptors and regulators have shown promise in pre-clinical and clinical testing, but most focus on other disease states, such as NAFLD or diabetes and not ALD. A number of bile acid-based therapies, including FXR agonists, TGR5 agonists, bile acid transporter inhibitors, and others, have been developed and show promise for the treatment of non-alcoholic steatohepatitis [144]. Currently, there are four pharmacotherapeutics in either phase 2 or 3 clinical trials for the treatment of alcohol-associated liver disease, with most specifically targeting AH. Larsucosterol (a DNA methyltransferase inhibitor) is recruiting for a phase 2b study, Canakinumab (an anti-IL-1β monoclonal antibody) is in phase 2 trials, Filgrastim (a granulocyte colony-stimulating factor) is recruiting for phase 3 trials, and Anakinra (an anti-IL1 receptor monoclonal antibody) plus zinc is recruiting for phase 2 trials. ALD prevalence is predicted to continue to increase and to result in more than one million deaths from 2019 to 2040 [25]. ALD has shown a significant association with changes in BA metabolism and homeostasis, with increased BA serum and hepatic levels in clinical patients. Overexpression of FGF19 and both systemic and intestinal-specific activation of FXR have been shown to ameliorate hepatic steatosis and inflammation in ethanol-fed mice [144]. ALD also causes a change in the gut microbiome composition, which can further impact BA metabolism and BA-mediated toxic effects via microbially conjugated BAs [78]. BAs are not only important in nutrient absorption but have been shown to be important signaling hormones, regulating lipid and glucose metabolism, cell proliferation, and inflammation [17,133,145,146]. There are currently several clinical trials pertaining to BAs and their role in disease modulation or as biomarkers for detection. In terms of trials targeting BAs or their receptors, there is currently a recruiting phase 3 clinical trial (ClinicalTrials.gov Identifier: NCT04956328) investigating obeticholic acid, a farnesoid X receptor agonist, and its effects on liver function in patients with primary biliary cirrhosis. Additionally, a phase 2 clinical trial using obeticholic acid showed a significant improvement in primary bile acid diarrhea but not in secondary bile acid diarrhea [147]. An actively recruiting phase 1/2 clinical trial (ClinicalTrials.gov Identifier: NCT03423121) aims to identify the safety and tolerability of bile acid supplementation in patients with progressive multiple sclerosis. Another actively recruiting clinical trial (ClinicalTrials.gov Identifier: NCT01200082) aims to evaluate the efficacy of monitoring sulfation of bile acids as a biomarker for hepatobiliary diseases. BAs play a pivotal role in the development of ALD and are ideal targets for the development of targeted therapeutics to combat ALD. Further research into their relationship is warranted to elucidate their interactions and identify pharmacological treatments for ALD.

## Figures and Tables

**Figure 1 cells-11-01374-f001:**
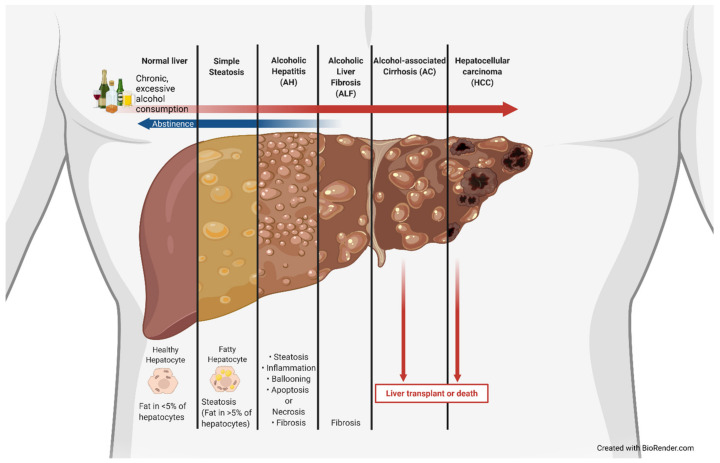
Alcohol-associated liver disease (ALD) spectrum. ALD spectrum showing the progression of pathology and contributing factors that initiate disease onset and promote disease progression. ALD is characterized by chronic and/or excessive alcohol intake, which leads to steatosis, inflammation, and fibrosis, culminating in cirrhosis and potential development of hepatocellular carcinoma. ALD is irreversible once the liver becomes cirrhotic.

**Figure 3 cells-11-01374-f003:**
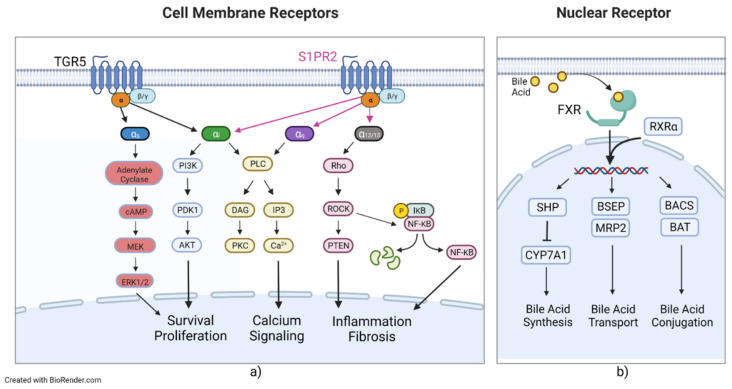
Bile acid signaling, surface-localized and nuclear-localized. (**a**) Graphical representation of cell membrane receptors for bile acids, highlighting signaling through the Takeda G protein-coupled receptor 5 (TGR5), sphingosine-1-phosphate receptor 2 (S1PR2), and G protein-coupled receptors. TGR5 can be associated with either the α_s_ or α_i_ subunit, whereas S1PR2 can associate with α_i_, α_q_, or α_12/13_. Activation of these receptors by binding of bile acids leads to several downstream effects. (**b**) Visualization of the farnesoid X receptor (FXR), a bile acid nuclear receptor. Activation of FXR by bile acid binding leads to heterodimerization with retinoid X receptor alpha (RXRα) and the transcriptional changes.

**Figure 4 cells-11-01374-f004:**
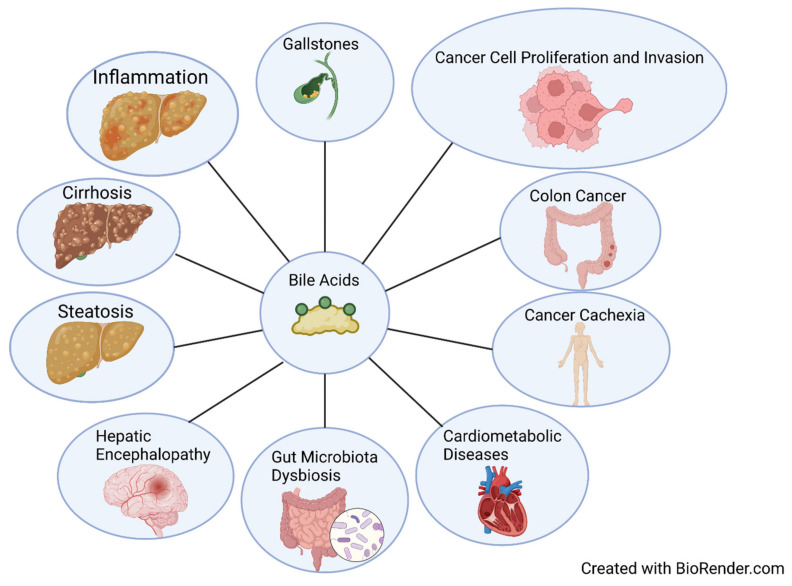
Bile acids in disease. Visual representation of the clinical significance of bile acids by showing their connection to different disease states and processes.

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
