# Peer review of "Key Signaling in Alcohol-Associated Liver Disease: The Role of Bile Acids"

_cells, 2022, doi:10.3390/cells11081374_

Round 1
Reviewer 1 Report
This is a review focusing on bile acids in patients with alcohol-associated liver disease. I have several comments.
- I would like to know the difference with other types of liver failure.
- What is the clinical significance of bile acid metabolism? Is there clinical trials focusing on bile acids?
Author Response
RESPONSE TO REVEIWERS’ COMMENTS
First, we would like to thank all three reviewers for their time spent reviewing our manuscript and for their helpful comments. We have made several changes in the revised manuscript based on the reviewers’ suggestions.
Reviewer 1
We would like to thank the reviewer for their helpful comments. The responses to each comment are listed below.
Comment: I would like to know the difference with other types of liver failure.
Response: We would like to thank the reviewer for this comment; however, we feel that the scope of this review is to focus on alcohol associated liver disease. Therefore, we have focused our manuscript to reflect that, although there is mention of and comparisons made to non-alcoholic fatty liver disease (NAFLD).
Comment: What is the clinical significance of bile acid metabolism? Is there clinical trials focusing on bile acids?
Response: We appreciate this insightful comment from the reviewer and have made changes to the document to reflect that. Bile acids are important signaling molecules. Disruption of bile acid metabolism has been implicated in various liver diseases including ALD, NAFLD, PSC and colorectal cancer. Clinical significance of BAs can be seen in section 3.4. There is currently a recruiting phase 3 clinical trial (ClinicalTrials.gov Identifier: NCT04956328) investigating obeticholic acid, a farnesoid X receptor agonist, effects on liver function in patients with primary biliary cirrhosis. As well as a phase 2 clinical trial using obeticholic acid that showed a significant improvement in primary bile acid diarrhea but not in secondary bile acid diarrhea (Walters et al., 2015).
Walters, J. R. F., Johnston, I. M., Nolan, J. D., Vassie, C., Pruzanski, M. E., & Shapiro, D. A. (2015). The response of patients with bile acid diarrhoea to the farnesoid X receptor agonist obeticholic acid. Alimentary Pharmacology & Therapeutics, 41(1), 54-64. https://doi.org/10.1111/apt.12999
Both clinical trial references have been added to the manuscript in the conclusion section.
Reviewer 2 Report
In this review Way et al describe the mechanisms involved in alcohol-associated liver disease and the associated role of bile acids. ADL is still of major health concern and therefore the topic of this paper is very relevant. The paper fits well in this special issue of "Cells". The manuscript is very well written and summarizes relevant scientific information in a very didactic way.
Minor comments:
- The review is based on chronic alcohol consumption, and besides the first sentence of the abstract, no reference is made to acute (and even acute-on-chronic) alcohol consumption. Are the effects of acute alcohol consumption on BA function/metabolism… different than those induced by chronic alcohol consumption? Can this be explored/elucidated in an additional paragraph.
- Line 89: for full completeness of the ethanol metabolism reactions, add reaction products of the conversion of acetaldehyde by ALDH.
- Line 178: sentence is not clear and needs rephrasing.
- Line 192: only uptake transporters are mentioned; efflux transporters (to bile canaliculi) could be added to this paragraph as well.
- Line 385: A figure summarizing the main factors discussed in this chapter (which is the most innovative/relevant of the review) would improve the manuscript considerably.
Author Response
RESPONSE TO REVEIWERS’ COMMENTS
First, we would like to thank all three reviewers for their time spent reviewing our manuscript and for their helpful comments. We have made several changes in the revised manuscript based on the reviewers’ suggestions.
Reviewer 2
We would like to thank reviewer 2 for their time and their thoughtful comments. Each comment made is addressed below.
Minor comments: The review is based on chronic alcohol consumption, and besides the first sentence of the abstract, no reference is made to acute (and even acute-on-chronic) alcohol consumption. Are the effects of acute alcohol consumption on BA function/metabolism… different than those induced by chronic alcohol consumption? Can this be explored/elucidated in an additional paragraph.
Response: The mentioning of acute was removed to reflect the scope of the review more accurately
Comment: Line 89: for full completeness of the ethanol metabolism reactions, add reaction products of the conversion of acetaldehyde by ALDH.
Response: The reaction products of aldehyde conversion of acetaldehyde were added (acetate and NADH).
Comment: Line 178: sentence is not clear and needs rephrasing.
Response: The sentence was broken up into smaller, clearer sentences.
Comment: Line 192: only uptake transporters are mentioned; efflux transporters (to bile canaliculi) could be added to this paragraph as well.
Response: We thank the reviewer for this insight and have added mention of the efflux transporters (BSEP and MRP2) to the manuscript at line 185.
Comment: Line 385: A figure summarizing the main factors discussed in this chapter (which is the most innovative/relevant of the review) would improve the manuscript considerably.
Response: We thank reviewer 2 for this suggestion. A new figure was made to visualize the relationship between bile acids and various diseases and altered states.
Reviewer 3 Report
Way and colleagues summarized recent knowledge about the contribution of bile acids to alcoholic liver disease. I have few suggestions which help to further improve the manuscript:
Line 13: What is meant by “acute alcohol use”? One might get the impression that even the consumption of alcohol for one day triggers ALD. The term “alcohol abuse” might be better here.
Line 68: “ALD is irreversible once the liver become significantly fibrotic.” This sentence is very vague... Of note, liver fibrosis is reversible under certain circumstances (see below).
Lines 52-53: “more deregulation of gene expression” This is also not very precise and must be specified.
Figure 1: The hepatocyte shown for "simple steatosis" looks like an apoptotic cell. Is this meant here? Lipid inclusions look a bit different.
Figure 1: ECM deposition contributes to liver regeneration and is not negative per se. Moreover, it is a highly dynamic process that is reversible under certain circumstances. You may draw the blue arrow from the center of the column “alcoholic liver fibrosis” to the left-hand side.
Figure 1: Normal liver instead of healthy liver.
Line 81 vs. 113: Different numbers are given here (?). Is it necessary to mention this again?
Line 138: The abbreviations SREB-1c and ChREBP must be explained.
Line 150: The paragraphs are very long. Please, add more paragraphs which will improve the readability. For instance at this position.
Line 151: Genetic was used twice in this sentence.
Line 174: Liver cirrhosis also affects the bone morrow.
Line 179: begins (?)
Figure 2: In the lower field of figure 2: Oxidation, Epimerization, Dehydroxylation, amino acid conjugation. Dehydoxylation but not Dehydrogenation.
Line 262: 27-hydroxycholesterol (typo)
Line 344: A citation is missing here.
Line 360: Observed is used twice in this sentence.
Author Response
RESPONSE TO REVEIWERS’ COMMENTS
First, we would like to thank all three reviewers for their time spent reviewing our manuscript and for their helpful comments. We have made several changes in the revised manuscript based on the reviewers’ suggestions.
Reviewer 3
We would like to thank reviewer 3 for their time and their very structured, insightful, and helpful comments. Each comment is addressed in the same format as submitted below.
Comment: Line 13: What is meant by “acute alcohol use”? One might get the impression that even the consumption of alcohol for one day triggers ALD. The term “alcohol abuse” might be better here.
Response: Acute was removed to reflect the scope of the paper more accurately
Comment: Line 68: “ALD is irreversible once the liver become significantly fibrotic.” This sentence is very vague... Of note, liver fibrosis is reversible under certain circumstances (see below).
Response: We would like to thank the reviewer for pointing this out. We made the changes as following “ALD is irreversible once the liver becomes cirrhotic.”
Comment: Lines 52-53: “more deregulation of gene expression” This is also not very precise and must be specified.
Response: The wording was changed to be more precise.
Comment: Figure 1: The hepatocyte shown for "simple steatosis" looks like an apoptotic cell. Is this meant here? Lipid inclusions look a bit different.\
Response: Image was changed to look less apoptotic.
Comment: Figure 1: ECM deposition contributes to liver regeneration and is not negative per se. Moreover, it is a highly dynamic process that is reversible under certain circumstances. You may draw the blue arrow from the center of the column “alcoholic liver fibrosis” to the left-hand side.
Response: Changed
Comment: Figure 1: Normal liver instead of healthy liver.
Response: Changed
Comment: Line 81 vs. 113: Different numbers are given here (?). Is it necessary to mention this again?
Response: The first number is specific to ALD (18,164/yr) the second number encompasses all chronic alcohol conditions that play a role in death (51,078/yr). Agree that it is probably not needed so it has been removed.
Comment: Line 138: The abbreviations SREB-1c and ChREBP must be explained.
Response: Acronyms defined
Comment: Line 150: The paragraphs are very long. Please, add more paragraphs which will improve the readability. For instance at this position.
Response: Paragraphs were split at the suggested position and at 246 and 327.
Comment: Line 151: Genetic was used twice in this sentence.
Response: one was removed
Comment: Line 174: Liver cirrhosis also affects the bone morrow.
Response: This line was summarizing findings found in reference 57. No mention was made in their study with the involvement of bone marrow, so it has not been added to this line. A reference was added to reflect the liver cirrhosis and bone marrow relationship after the specified sentence.
Comment: Line 179: begins (?)
Response: Yes, it was typo, changed to begins.
Comment: Figure 2: In the lower field of figure 2: Oxidation, Epimerization, Dehydroxylation, amino acid conjugation. Dehydroxylation but not Dehydrogenation.
Response: Changed
Comment: Line 262: 27-hydroxycholesterol (typo)
Response: Spelling mistake corrected
Comment: Line 344: A citation is missing here.
Response: Citation added.
Comment: Line 360: Observed is used twice in this sentence.
Response: Observed changed to revealed
Round 2
Reviewer 1 Report
The authors have revised the manuscript appropriately.
Author Response
We would like to thank the reviewer for their time again. No further revisions were suggested so no changes were made.